# DebiasDiff: Debiasing Text-to-image Diffusion Models with Self-discovering Latent Attribute Directions

## Abstract

While Diffusion Models (DM) exhibit remarkable performance across various image generative tasks, they nonetheless reflect the inherent bias presented in the training set. As DMs are now widely used in real-world applications, these biases could perpetuate a distorted worldview and hinder opportunities for minority groups. Existing methods on debiasing DMs usually requires model re-training with a human-crafted reference dataset or additional classifiers, which suffer from two major limitations: (1) collecting reference datasets causes expensive annotation cost; (2) the debiasing performance is heavily constrained by the quality of the reference dataset or the additional classifier. To address the above limitations, we propose DebiasDiff, a plug-and-play method that learns attribute latent directions in a self-discovering manner, thus eliminating the reliance on such reference dataset. Specifically, DebiasDiff consists of two parts: a set of attribute adapters and a distribution indicator. Each adapter in the set aims to learn an attribute latent direction, and is optimized via noise composition through a self-discovering process. Then, the distribution indicator is multiplied by the set of adapters to guide the generation process towards the prescribed distribution. Our method enables debiasing multiple attributes in DMs simultaneously, while remaining lightweight and easily integrable with other DMs, eliminating the need for re-training. Extensive experiments on debiasing gender, racial, and their intersectional biases show that our method outperforms previous SOTA by a large margin.

## 1 Introduction

State-of-the-art Text-to-Image Diffusion Models (DMs) such as Stable Diffusion (Rombach et al., 2022), DALL-E 3 (Ramesh et al., 2022) and Imagen (Saharia et al., 2022) have demonstrated remarkable performance in generating high-quality images. With the rapid development of DMs, an increasing number of individuals and corporations are choosing to utilize them to serve their own purposes. For instance, Stable Diffusion v1.5 has been downloaded over 8 million times on the Huggingface repository, and Midjourney is used by over a million users (Fatunde & Tse, 2022). However, existing DMs have been found to generate biased content across various demographic factors, such as gender and race (Luccioni et al., 2023), which could have harmful effects on society when these models are implemented in real-world applications.

In Figure 1, we randomly generate several images of four occupations using Stable Diffusion v2.1. Given the prompt of 'A photo of a CEO' or 'A photo of a doctor', the generated images predominantly depict male figures, reinforcing the stereotype that leadership roles and highly respected professions, such as CEOs and doctors, are male-dominated. On the contrary, when the prompt is 'A photo of an executive assistant' or 'A photo of a nurse', the majority of generated images depict female figures, reflecting the bias that administrative or supportive roles are traditionally associated with women. Regarding racial bias, we randomly generate 1000 images using Stable Diffusion v2.1 with the prompt 'A photo of a worker'. The statistic of 1000 images depicted in Figure 2 shows a strong bias in racial representation, with White individuals making up 71% of the total, while minority groups like Middle Eastern, Latino, Black, and Indian each account for only 3-4%. This bias in DM, produce less accurate or fair results for underrepresented populations. We further investigate such bias situation across across different versions of DMs. We randomly generate 1000

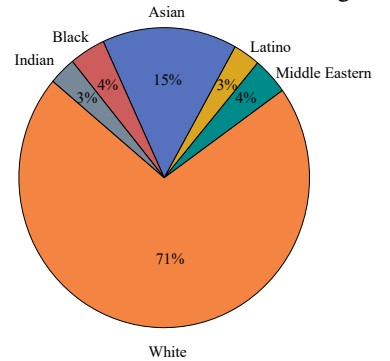
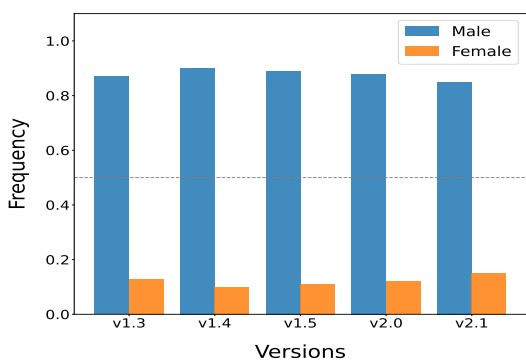

(a) Prompt: 'A photo of a CEO'

(b) Prompt: 'A photo of a doctor'

(c) Prompt: 'A photo of an executive assistant'

(d) Prompt: 'A photo of nurse'

Figure 1: Illustration of gender bias associated with different occupations in Stable Diffusion v2.1. The leadership roles and respected professions such as CEOs and doctors are biased towards male figures, whereas administrative or supportive roles such as executive assistant and nurses are biased towards female figures.

Figure 2: Racial bias in randomly generated 1000 images with the prompt 'A photo of a worker' using Stable Diffusion v2.1.

Figure 3: Gender bias in randomly generated 1000 images with the prompt 'A photo of a CEO' using Stable Diffusion v1.3, v1.4, v1.5, v2.0, and v2.1.

images with the prompt of 'A photo of a CEO' using five versions of DMs: Stable Diffusion v1.3 (Rombach, 2022a), v1.4 (Rombach, 2022b), v.1.5 (Rombach, 2022c), v2.0 (Rombach, 2022d) and v2.1 (Rombach, 2022e). Figure 3 demonstrates that gender bias exists across all versions of DMs.

This bias arises from two main factors. First, the training data sourced from the web is inherently biased. Second, the bias is partly inherited from the CLIP (Radford et al., 2021) model used in the generation process. Biases in the generated data are even more pronounced in large text-to-image DMs, where models often produce content that associates specific genders with particular professions. Several work has attempted to mitigate the bias in DMs by re-training the model (Shen et al., 2024) or training-free approaches (Parihar et al., 2024; Gandikota et al., 2023; Orgad et al., 2023). In training-based methods, Shen et al. (2024) propose to manually curate a reference dataset and match the distribution of generated images with that of the reference dataset. Among training-free approaches, Parihar et al. (2024) exploit the rich demographic information embedded in the latent features of the denoising U-Net to guide the generation. Although this method avoids re-training DMs, it depends on training an MLP-based Attribute Distribution Predictor using pseudo labels generated from existing attribute classifiers. This reliance on accurate attribute classifiers for training $h$-space classifiers significantly limits its debiasing performance. Additionally, Gandikota et al. (2023) and Orgad et al. (2023) employ closed-form editing to adjust concepts within DMs without re-training. However, training-based methods heavily depend on gathering annotated reference datasets, which are both expensive and constrained by the dataset's quality. Despite the implementation efficiency of training-free methods, they tend to be less effective than training-based approaches.

To address these limitations, we propose DebiasDiff, a plug-and-play method that automatically learns attribute latent directions, removing the dependency on reference datasets. DebiasDiff is composed of two components: a set of attribute adapters and a distribution indicator. Each adapter is trained to learn an attribute-specific latent direction, optimized through noise composition in a self-discovering manner, i.e., our method automatically learns attribute latent directions without relying on a labeled reference dataset. Through noise composition, our method explores and optimizes attribute directions directly from the model's latent space, uncovering patterns without external supervision. At inference stage, the distribution indicator is then applied to select attribute-specific

adapters, guiding the generative process towards the desired distribution. We comprehensively evaluate the effectiveness of our approach in debiasing gender, racial, and intersectional biases using occupational prompts. Experimental results demonstrate that our method not only achieves state-of-the-art performance in single and multiple attribute debiasing tasks but also preserves the generation quality of DM. Furthermore, we show that once DebiasDiff is trained on one diffusion model, it can be seamlessly integrated into other models without re-tuning. Thanks to its strong transferability and plug-and-play functionality, our method offers a practical solution for both individual users and organizations, facilitating the responsible use of diffusion models in future applications.

To summarize, our main contributions are as following:

- We propose DebiasDiff, a novel method for debiasing DMs by learning attribute latent directions in a self-discovering manner, eliminating the reliance on the reference dataset or classifier, and thus significantly reduce the cost.

- DebiasDiff is lightweight, plug-and-play and shows good transferability across different DMs, making it more convenient to deploy in the real world.

- Extensive experiments show that our method achieves SOTA performance across diverse debiasing tasks while retaining the image generation quality.

## 2 RELATED WORK

**Bias in Diffusion Models.** Diffusion models for text-to-image generation (T2I) have been observed to produce biased and stereotypical images, even when given neutral prompts. Cho et al. (2023) found that Stable Diffusion (SD) tends to generate images of males when prompted with occupations, with skin tones predominantly centered around a few shades from the Monk Skin Tone Scale (Monk, 2023). Seshadri et al. (2023) noted that SD reinforces gender-occupation biases present in its training data. In addition to occupations, Bianchi et al. (2023) discovered that simple prompts involving character traits and other descriptors also result in stereotypical images. Luccioni et al. (2023) created a tool to compare generated image collections across different genders and ethnicities. Moreover, Wang et al. (2023) introduced a text-to-image association test and found that SD tends to associate females more with family roles and males more with career-related roles.

**Debiasing Diffusion Models by retraining.** Before DMs, previous approaches mainly focus on debiasing GAN models by assuming access to the labels of sensitive attributes and aim to debias the models, ensuring no correlation exists between the decision attribute and the sensitive attribute. (Nam et al., 2023; Xu et al., 2018; van Breugel et al., 2021; Sattigeri et al., 2019; Yu et al., 2020; Choi et al., 2020; Teo et al., 2023; Um & Suh, 2023). More recently, regarding DMs, Shen et al. (2024) propose a distributional alignment loss to guide the characteristics of the generated images towards target distribution and use adjusted direct finetuning to directly optimize losses on the generated images. Their method requires a reference training dataset to complete the retraining process, whereas our method does not need such reference dataset, which largely reduce annotation costs.

**Debiasing Diffusion Models without training.** Parihar et al. (2024) propose Distribution Guidance (DG), which guides the generated images to follow the prescribed attribute distribution. Although DG does not require retraining of DMs, it requires training an Attribute Distribution Predictor (ADP), which is a small MLP that maps the latent features to the distribution of attributes. Since ADP is trained with pseudo labels generated from existing attribute classifiers, the performance of DG is largely constrained by the accuracy of attribute classifiers. Gandikota et al. (2023) and Orgad et al. (2023) use closed-form editing approach to edits concepts inside DM without training. Despite being easy to implement, its effectiveness is weaker compared with training-based approach.

## 3 PRELIMINARY

**Latent Diffusion Models (LDMs)** (Rombach et al., 2022), also known as Stable Diffusion (SD), perform the diffusion process within the latent space. During training, noise is added to the encoded latent representation of the input image $x$, resulting in a noisy latent code $z_t$ at each time step $t$.

In the pretraining stage, an autoencoder framework is employed to map images into a lower-dimensional latent space via an encoder: $z = \mathcal{E}(x)$. The decoder then reconstructs images from these latent codes: $x \approx \mathcal{D}(\mathcal{E}(x))$. This process ensures that the latent space retains the essential semantic information of the image.

Figure 4: Overview of DebiasDiff's training pipeline. Attribute-specific adapters ($M$) are attached to the cross-attention layers in the denoising UNet. Target group and attribution direction are fed into the DM for composing the noise predictions (Eq. 5), which is used as self-discovering attribute direction guidance to optimize the adapters.

The training objective of the diffusion model in the latent space is given by:

$$\mathcal{L}_{\text{LDM}} = \mathbb{E}_{z \sim \mathcal{E}(x), c, \epsilon \sim \mathcal{N}(0,1), t} \left[ \left\| \epsilon - \epsilon_\theta \left( z_t, c, t \right) \right\|_2^2 \right], \quad (1)$$

where $\epsilon$ is Gaussian noise sampled from a normal distribution $\mathcal{N}(0, 1)$, $\epsilon_\theta$ is the denoising network, and $c$ represents any conditioning embeddings (e.g., text or class labels).

At the inference stage, a latent code $z_T$ is sampled from Gaussian noise at the initial timestep $T$. The denoising network $\epsilon_\theta$ is then applied iteratively to remove the noise over several steps, generating a denoised latent representation $z_0$. Finally, the pretrained decoder reconstructs the image from the denoised latent code: $\hat{x}_0 \approx \mathcal{D}(z_0)$, where $\hat{x}_0$ is the generated output image.

**Classifier-free Guidance** (Ho & Salimans, 2022) aim to modulate image generation by steering the probability distribution towards data that is more probable according to an implicit classifier $p(c \mid z_t)$. It operates at inference phase and the model is jointly trained on both conditional and unconditional denoising tasks. During inference, both the conditional and unconditional denoising scores are derived from the model. The final score $\tilde{\epsilon}_\theta(z_t, c, t)$ is then adjusted by weighting the conditioned score more heavily relative to the unconditioned score using a guidance scale $\alpha > 1$.

$$\tilde{\epsilon}_\theta(z_t, c, t) = \epsilon_\theta(z_t, t) + \alpha(\epsilon_\theta(z_t, c, t) - \epsilon_\theta(z_t, t)) \quad (2)$$

The inference process begins with sampling a latent variable $z_T \sim \mathcal{N}(0, 1)$, which is subsequently denoised using $\tilde{\epsilon}_\theta(z_t, c, t)$ to obtain $z_{t-1}$. The denoising is performed iteratively until obtaining $z_0$. Finally, the decoder transforms the latent representation $z_0$ back into image space: $x_0 \leftarrow \mathcal{D}(z_0)$.

## 4 METHOD

Given a Diffusion Model, we aim at reducing the bias in the DM by attaching and learning a set of light-weight adapters, each of which represents a category of an attribute (e.g., female of gender), guiding the DM towards an attribute latent direction. Unlike previous work that relies on additional reference datasets and has to finetune the whole DM (Shen et al., 2024), we instead optimize the attached adapters via noise composition through a self-discovering process (detailed in Section 4.2). This significantly reduces the cost in computation and data. During the inference stage, given a predefined target distribution (e.g., uniform), we introduce a distribution indicator implemented by a gating function to select one corresponding adapter which will be attached to the DM for generating image. In this way, the set of generated images will follow the predefined target distribution, and thus are not biased to some categories of an attribute (if the predefined target distribution is uniform). Our training and inference diagrams are illustrated in Figure 4 and Figure 5. In the following sections, we start elaborating our method in single attribute settings and then extend it to more general ones.

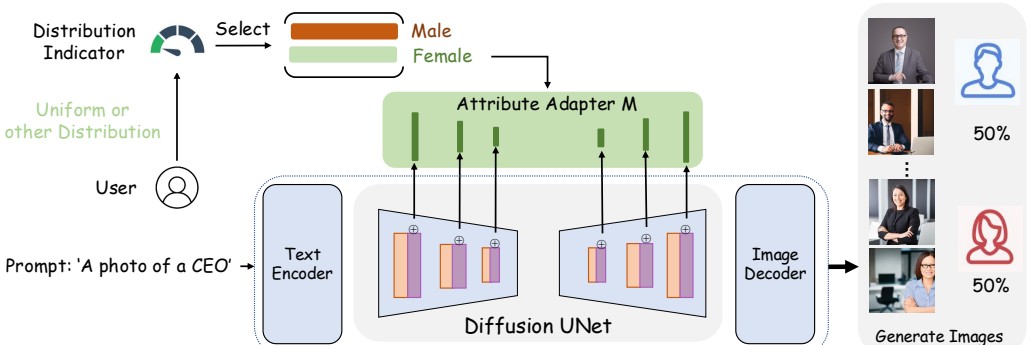

Figure 5: Overview of DebiasDiff's inference pipeline. The distribution indicator is generated according the prescribed distribution. Then, it is multiplied by the set of attribute adapters matrices to select attribute matrix adapter. The selected adapter is integrated to the DM with no overhead, guiding the generation towards the prescribed distribution.

## 4.1 ATTACHING LIGHT-WEIGHT ATTRIBUTE-SPECIFIC ADAPTERS

To debias a DM for generating images of a given attribute that contains several categories (e.g., male and female are two categories of gender attribute), we first aim at equip the DM with skills of generating images for each category. To achieve this, we attach a light-weight adapter per category in each layer of the DM inspired by parameter-efficient fine-tuning (PEFT) instead of finetuning the whole model to acheive a good trade-off between performance and computational cost. In this work, we use the 1-dim adapter (Lyu et al., 2024) and only add the adapter to each cross-attention layers of the denoising U-Net, as shown in Figure 4 as we find that attaching adapters to all layers does not help and will also increase the computational cost.

Specifically, for the $i$-th cross-attention layer parameterized by $\boldsymbol{W}_i \in \mathbb{R}^{m \times n}$ in the denoising U-Net, we attach an adapter to the layer to guide the attribute towards a certain category. The adapter consists of two 1-dim vectors: $\boldsymbol{p} \in \mathbb{R}^m$ and $\boldsymbol{q} \in \mathbb{R}^n$. The forward process of the $i$-th cross-attention layer would be updated from $\boldsymbol{y}_i = \boldsymbol{W}_i \boldsymbol{x}_i$ to $\boldsymbol{y}_i = \boldsymbol{W}_i \boldsymbol{x}_i + (\boldsymbol{q}_i^T \boldsymbol{x}_i) \cdot \boldsymbol{p}_i$. $\boldsymbol{x}_i \in \mathbb{R}^n$ and $\boldsymbol{y}_i \in \mathbb{R}^m$ represent the input and output of the layer, and superscript $T$ indicates transposition. Thus, adapters in all ($r$) cross-attention layers for an attribute category $d$ are:

$$M_d = \boldsymbol{Q}^T \boldsymbol{P}, \tag{3}$$

where $\boldsymbol{Q} = [\boldsymbol{q}_1, \boldsymbol{q}_2, \ldots, \boldsymbol{q}_r], \boldsymbol{P} = [\boldsymbol{p}_1, \boldsymbol{p}_2, \ldots, \boldsymbol{p}_r]$. Each column vector in $\boldsymbol{Q}$ and $\boldsymbol{P}$, we pad 0 to their end if their dimensions are not the same.

## 4.2 OPTIMIZING ADAPTERS VIA SELF-DISCOVERING PROCESS

One straightforward way to optimize the attached adapters is to collect a unbiased reference dataset as in prior work (Shen et al., 2024). However, it is expensive to collect such dataset and the quality of the dataset would also limites the performance. To this end, we propose to train the adapters in a self-discovering manner.

Given a target group $g_t$ (for example, 'CEO') and model $\theta$, we want to optimize the adapters such that the model attached with the adapters generate image $X$ towards certain attribute category $d$ (for example, 'male' or 'female' ) when conditioned on $g_t$:

$$P_{\theta^*}(X|g_t) \leftarrow P_\theta(X|g_t) \left(P_\theta(d|X)\right)^\eta, \tag{4}$$

where $P_\theta(X|g_t)$ represents the distribution generated by the original model when conditioned on $g_t$, and $\theta^*$ represents the new model equipped with the adapters of $d$. Note that $g_t$ can be set to an empty string '' so that the model will be debiased for all possible groups.

Applying the Bayes Formula, $P(d|X) = \frac{P(X|d)P(d)}{P(X)}$ to Eq. 4, taking logarithm on both sides, we are able to derive that the gradient of the log probability $\nabla \log P_{\theta^*}(X|g_t)$ would be proportional to:

$$\nabla \log P_\theta(X|g_t) + \eta \left(\nabla \log P_\theta(X|d) - \nabla \log P_\theta(X|g_t)\right) \tag{5}$$

Based on Tweedie's formula (Efron., 2011) and the reparametrization trick of Classifier-free guidance (Ho & Salimans, 2022), we introduce a time-varying noising process and represent each score (gradient of log probability) as a denoising prediction $\epsilon(X_t, c_t, t)$, which leads to our learning objective for adapters of category $d$ of an attribute.

$$\epsilon_{\theta^*}(X_t, g_t, t) \leftarrow \epsilon_\theta(X_t, g_t, t) + \eta \left( \epsilon_\theta(X, d, t) - \epsilon_\theta(X_t, g_t, t) \right) \tag{6}$$

Therefore, the guidance loss for optimizing adapters of category $d$ can be defined as follows:

$$\mathcal{L}_{\text{Guidance}} = \mathbb{L}_{x_t, t} \left[ \|\epsilon_{\theta^*}(X_t, g_t, t) - \epsilon_L\|^2 \right],$$
$$\epsilon_L = \epsilon_\theta(X_t, g_t, t) + \eta \left( \epsilon_\theta(X_t, d, t) - \epsilon_\theta(X_t, g_t, t) \right) \tag{7}$$

where the goal is to align the noise $\epsilon_{\theta^*}(X_t, g_t, t)$ of the $\theta^*$ (DM with adapters of category $d$) and the noise composition $\epsilon_L$ of group $g_t$ and category $d$ from the frozen DM $\theta$. And hence, our adapters optimization does not require any additional data.

## 4.3 INFERENCE WITH DISTRIBUTION INDICATOR.

After the optimization, we obtain a set of adapters $\mathcal{M} = \{\boldsymbol{M}_1, \boldsymbol{M}_2, \ldots, \boldsymbol{M}_t\}$ for $t$ categories of a given attribute. For instance, $t = 2$ for gender bias and $t = 4$ for racial bias in our case. As shown in Figure 5, at the inference stage, we introduce a distribution indicator $\boldsymbol{h}$. Given a prescribed distribution $f_\theta^a$ (e.g., uniform distribution), we define its Probabilistic Mass Function (PMF) as follows:

$$P(X = x) = f_\theta^a(x) = \begin{cases} \frac{1}{t} & \text{for } x \in \{1, 2, \ldots, t\}, \\ 0 & \text{otherwise}, \end{cases} \tag{8}$$

where $t$ is the number of possible categories for each attribute, and $\{1, 2, \ldots, t\}$ represents the set of all possible values for the random variable $X$. The parameter $\theta$ controls the shape of the prescribed distribution, and in the case of the uniform distribution, $\theta$ implies that all outcomes in the set have equal probability, i.e., $\frac{1}{t}$.

Then, we randomly sample an index $k$ from the prescribed distribution $f_\theta^a$. The distribution indicator $\boldsymbol{h} \in \mathbb{R}^t$ is formulated to reflect the chosen index $k$ as follows:

$$\boldsymbol{h}_i := \begin{cases} 1 & \text{if } i = k, \\ 0 & \text{if } i \neq k, \end{cases} \tag{9}$$

where $i \in \{1, 2, \ldots, t\}$, and $k$ is the sampled index based on the distribution $f_\theta^a$. The indicator $\boldsymbol{h}$ is a one-hot vector, where the $k$-th element is 1, indicating the sampled outcome, and all other elements are 0. After obtaining the distribution indicator, it is multiplied by the set of trained attribute matrix adapters. Subsequently, the final weight change $\Delta \boldsymbol{W}$ is given by:

$$\Delta \boldsymbol{W} = \boldsymbol{h} \cdot \mathcal{M}. \tag{10}$$

And the model is updated as $\boldsymbol{W} \leftarrow \boldsymbol{W} + \alpha \Delta \boldsymbol{W}$, where $\alpha$ is a scaling factor controlling the strength of the guidance.

## 4.4 DEBIASING MULTIPLE ATTRIBUTES (INTERSECTIONAL DEBIASING)

Our method can be inherently extended to debiasing multiple attributes in diffusion models (DM). Specifically, in the case of multiple attribute debiasing, the adapter for each attribute should not interfere with the others. Otherwise, the most recently trained adapter could degrade the performance of previously learned adapters. For each attribute to be debiased, we denote a set of adapter parameters as $\{\boldsymbol{P}_t, \boldsymbol{Q}_t\}$. We have $\boldsymbol{P}_t = \left[\boldsymbol{p}_t^1, \boldsymbol{p}_t^2, \ldots, \boldsymbol{p}_t^r\right], \boldsymbol{Q}_t = \left[\boldsymbol{q}_t^1, \boldsymbol{q}_t^2, \ldots, \boldsymbol{q}_t^r\right]$.

To avoid interference between attribute adapters, we extend Eq. 7 by introducing an orthogonal regularization loss that regularizes the vector subspace spanned by each $\boldsymbol{P}_t$ and $\boldsymbol{Q}_t$ to be orthogonal to each other:

$$\mathcal{L}_{\text{orth}} = \sum_{i=1}^{t-1} \left( \boldsymbol{P}_i \times \boldsymbol{P}_t + \boldsymbol{Q}_i \times \boldsymbol{Q}_t \right). \tag{11}$$

$$\mathcal{L} = \mathcal{L}_{\text{Guidance}} + \gamma \mathcal{L}_{\text{orth}}. \tag{12}$$

Table 1: Comparisons of our method to the SOTA methods in gender bias over two predefined distributions: $f_\theta^1 = (0.5, 0.5)$ and $f_\theta^2 = (0.2, 0.8)$, representing the probability of male and female respectively.

| Method | FD ↓ | | CLIP$_{sim}$ ↑ | | BRISQUE ↑ | |
|--------|------|------|------|------|------|------|
| | $f_\theta^1$ | $f_\theta^2$ | $f_\theta^1$ | $f_\theta^2$ | $f_\theta^1$ | $f_\theta^2$ |
| Original SD | 0.424 | 0.847 | **0.38** | **0.38** | **38.65** | 38.69 |
| F4Fair | 0.165 | 0.387 | 0.36 | 0.35 | 38.21 | 37.64 |
| H Guidance | 0.118 | 0.398 | 0.31 | 0.32 | 38.54 | 38.65 |
| UCE | 0.284 | 0.536 | 0.36 | 0.29 | 37.12 | 36.54 |
| Ours | **0.003** | **0.005** | **0.38** | **0.38** | 38.46 | **38.72** |

Table 2: Comparisons of our method to the SOTA methods in racial bias over two distributions: $f_\theta^1 = (0.25, 0.25, 0.25, 0.25)$ and $f_\theta^2 = (0.4, 0.3, 0.2, 0.1)$, representing probability of WMELH, Asian, Black, and Indian respectively.

| Method | FD ↓ | | CLIP$_{sim}$ ↑ | | BRISQUE ↑ | |
|--------|------|------|------|------|------|------|
| | $f_\theta^1$ | $f_\theta^2$ | $f_\theta^1$ | $f_\theta^2$ | $f_\theta^1$ | $f_\theta^2$ |
| Original SD | 0.384 | 0.497 | **0.46** | **0.41** | 38.94 | 38.65 |
| F4Fair | 0.220 | 0.305 | 0.43 | 0.40 | 38.80 | 38.45 |
| H Guidance | 0.150 | 0.285 | 0.41 | 0.37 | 38.65 | 38.60 |
| UCE | 0.290 | 0.460 | 0.45 | 0.39 | 37.90 | 37.80 |
| Ours | **0.095** | **0.150** | **0.46** | **0.41** | **38.96** | **38.66** |

# 5 EXPERIMENT

In this section, we first describe the implementation details, evaluation metrics and baseline methods in this work, then present a quantitative and qualitative analysis of our method.

## 5.1 EXPERIMENTAL DETAILS

**Implementation details.** We use Stable Diffusion v2.1 for all methods. We employ the prompt template "*a photo of the face of a* {occupation}, *a person*". At inference time, for each bias, we generate 100 images per occupation across 100 occupations, resulting in a total of 10,000 images. We set $\eta = \alpha = 1$, and train for 1000 iterations with a learning rate of 1e-5. For gender bias, we use the CelebA (Liu et al., 2015) dataset to train a binary classifier with two categories:{male,female}. For racial bias, we use the FairFace (Joo, 2021) dataset to train a classifier with the following four categories: WMELH={White, Middle Eastern, Latino Hispanic}, Asian={East Asian, Southeast Asian}, Black, and Indian. Please refer to supplementary for more details.

**Compared methods.** In this work, we compare our method with recent state-of-the-art (SOTA) methods, including a retraining-based method, **Finetuning for Fairness (F4Fair)** (Shen et al., 2024), a training-free approach, **H-Distribution Guidance (H Guidance)** (Parihar et al., 2024), and a closed-form editing approach, **Unified Concept Editing (UCE)** (Gandikota et al., 2023). Please refer to Appendix A.2 for more detailed introduction of these methods.

## 5.2 EVALUATION METRICS

We evaluate the debias performance of all methods in three metrics:

**Fairness Discrepancy (FD).** Following prior work(Parihar et al., 2024), we adopt the Fairness Discrepancy (FD) metric. For an attribute $a$ and target distribution $p_\theta^a$, we use a high-accuracy classifier $\mathcal{C}_a$ to compute the fairness performance: $||p_\theta^a - \mathbb{E}_{\mathbf{x} \sim p_\theta(\mathbf{x})}(\mathbf{y})||_2$ where $\mathbf{y}$ is the softmax output of $\mathcal{C}_a(\mathbf{x})$. The target distribution $p_\theta^a$ can be any user-defined vector, typically uniform. A lower FD score indicates a closer match to the target distribution.

**CLIP$_{sim}$.** Besides fairness, the debiased model should maintain the ability to generate images that are semantically close to their text prompts. Therefore, following (Shen et al., 2024), we report the CLIP similarity score CLIP$_{sim}$ between the generated image and its prompt.

**BRISQUE.** The generated image quality is also important as the debiasing process shown not influence the image generation ability. Thus, we use the BRISQUE metric for evaluate the quality of the generated images as in the prior work (Parihar et al., 2024).

## 5.3 RESULTS

**Comparisons in gender debiasing.** Table 1 demonstrates that our method outperforms others in mitigating gender bias over two predefined distributions: $f_\theta^1 = (0.5, 0.5)$, representing equal likelihood of male and female, and $f_\theta^2 = (0.2, 0.8)$, where male and female have a 20% and 80% probability, respectively. Original SD model exhibits high FD scores (0.424 at $f_\theta^1$ and 0.847 at $f_\theta^2$), indicating significant bias towards one gender. While previous methods like F4Fair, H Guidance,

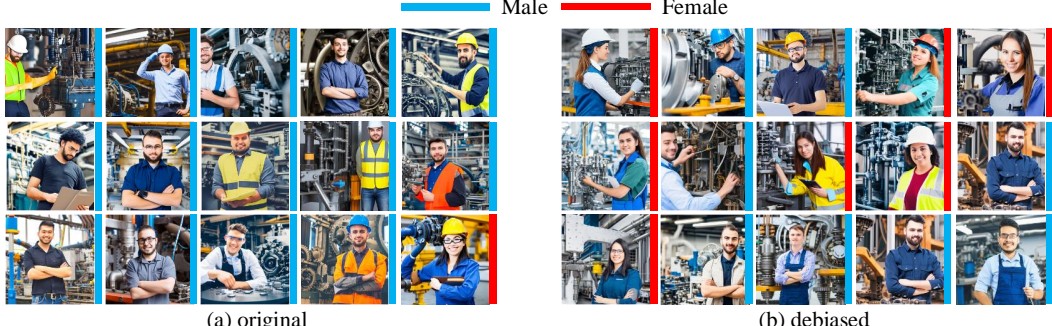

| | Male | | Female | |

(a) original                                    (b) debiased

Figure 6: Images generated from the original SD (left) and Ours for gender and race (right) with prompt 'A photo of a work'. Gendet ratio: Male : Female = $13 : 2 \rightarrow 8 : 7$

and UCE reduce bias to some extent, our method achieves the most reduction, with FD scores of 0.003 at $f_\theta^1$ and 0.005 at $f_\theta^2$. Additionally, our method preserves semantic similarity (CLIP$_{\text{sim}}$ of 0.38) and image quality (BRISQUE scores of 38.46 and 38.72), matching or slightly surpassing the Original SD model. The qualitative results in Figure 6 further demonstrate that our approach effectively mitigates gender bias without compromising image quality or semantic coherence.

**Comparisons in racial debiasing.** Table 2 compares methods over two distributions: $f_\theta^1 = (0.25, 0.25, 0.25, 0.25)$, representing equal probabilities for WMELH, Asian, Black, and Indian, and $f_\theta^2 = (0.4, 0.3, 0.2, 0.1)$, with higher probabilities for WMELH and Asian, and lower probabilities for Black and Indian. Original SD model shows significant racial bias, with FD scores of 0.384 for $f_\theta^1$ and 0.497 for $f_\theta^2$, indicating poor calibration to either distribution.

Debiasing methods such as F4Fair, H Guidance, and UCE reduce this bias, with H Guidance achieving relatively lower FD scores. However, our method performs the best, reducing bias to 0.095 for $f_\theta^1$ and 0.150 for $f_\theta^2$. In terms of semantic similarity, the Original SD model sets a strong baseline (0.46 for $f_\theta^1$ and 0.41 for $f_\theta^2$), and our method maintains this high alignment, ensuring that debiasing does not impair semantic accuracy. Regarding image quality, our approach slightly improves upon the Original SD model, achieving the highest scores (39.10 for $f_\theta^1$ and 38.90 for $f_\theta^2$). Again, the qualitative results in Figure 7 further verify that our method outperforms others in reducing racial bias while preserving both semantic similarity and image quality.

**Comparisons in intersectional debiasing** We also consider a more complex challenge of intersectional debiasing, i.e., jointly debiasing both gender and racial biases. The target distribution $f_\theta$ is set to be uniform for both gender and racial bias. Table 3 shows that Original SD model exhibits significant bias with an FD score of 0.214. While F4Fair and H Guidance reduce this bias to 0.145 and 0.130, respectively, our method

Table 3: Evaluation of mitigating intersectional bias across methods.

| Method | FD ↓ | CLIP$_{\text{sim}}$↑ | BRISQUE ↑ |
|---|---|---|---|
| Original SD | 0.214 | 0.35 | 39.24 |
| F4Fair | 0.145 | 0.36 | 38.90 |
| H Guidance | 0.130 | 0.33 | 38.75 |
| UCE | 0.180 | 0.34 | 38.60 |
| Ours | **0.047** | **0.36** | **39.50** |

achieves a much lower FD score of 0.047, reflecting a substantial improvement in fairness.

For semantic similarity, the Original SD scores 0.35, and F4Fair slightly improves it to 0.36. Our method matches this performance, maintaining semantic coherence while reducing bias. Regarding image quality, the Original SD has a BRISQUE score of 39.24, while our method improves it to 39.50, indicating enhanced perceptual quality. These results demonstrate that our method excels at jointly debiasing both gender and racial biases, significantly reducing bias without sacrificing semantic accuracy or image quality. This highlights its robustness and practicality in real-world settings where multiple biases are present.

**Transferability across different DMs.** We further evaluate the transferability of our method by training it with Stable Diffusion v2.1 and testing it over other versions, and results are reported in Table 4. From the table we can see that the performance of in all metrics are close to the optimal results achieved when training and testing are performed using the same model version (v2.1). While there is a slight increase in FD when tested on v1.4, v1.5, and v2.0, the differences are minor, indicating that our method can effectively generalize across different model versions. This robust-

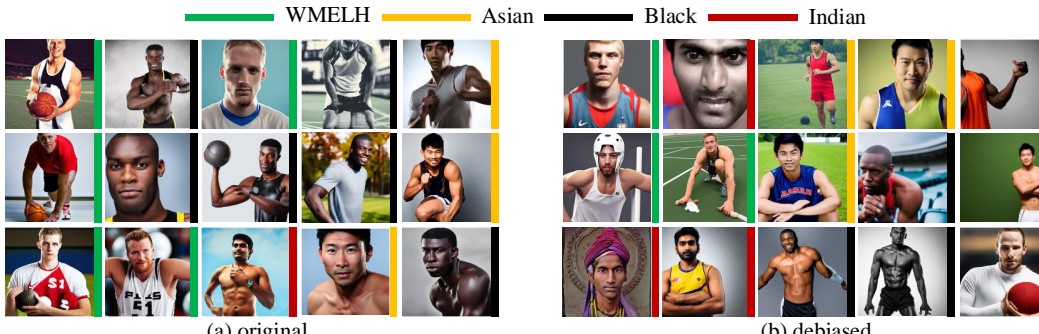

| | WMELH | Asian | Black | Indian |

(a) original         (b) debiased

Figure 7: Images generated from original SD (left) and Ours for gender and race (right) with prompt 'A photo of a sportsman'. Racial group distribution: WMELH : Asian : Black:Indian = 7:2:5:1 → 4:4:4:3

Table 4: Transferability of proposed method across different DMs

| Train | Test | Gender | | | Racial | | |
|---|---|---|---|---|---|---|---|
| | | FD $\downarrow$ | CLIP$_{sim}$ $\uparrow$ | BRISQUE $\uparrow$ | FD $\downarrow$ | CLIP$_{sim}$ $\uparrow$ | BRISQUE $\uparrow$ |
| v2.1 | v1.4 | 0.006 | 0.37 | 38.40 | 0.097 | 0.45 | 38.90 |
| v2.1 | v1.5 | 0.006 | 0.37 | 38.42 | 0.096 | 0.45 | 38.92 |
| v2.1 | v2.0 | 0.008 | 0.38 | 38.45 | 0.095 | 0.46 | 38.95 |
| v2.1 | v2.1 | **0.003** | **0.38** | **38.46** | **0.095** | **0.46** | **38.96** |

ness highlights the method's capability to transfer learned features across varying model conditions, underscoring its practical value for real-world applications.

## 5.4 ABLATION STUDY

**Attaching adapters in U-Net.** The results in Table 5 illustrate the impact of attaching adapters to different layers of the U-Net in the DM. When adapters are attached to all layers, the fairness score (FD) is 0.165, with a CLIP similarity score of 0.33 and a BRISQUE score of 35.00. The best results are obtained when

Table 5: Ablation of layers that are attached adapters.

| Location | FD $\downarrow$ | CLIP$_{sim}$ $\uparrow$ | BRISQUE $\uparrow$ |
|---|---|---|---|
| All layers | 0.165 | 0.33 | 35.00 |
| Non-CA layers | 0.158 | 0.34 | 37.00 |
| CA layers | **0.047** | **0.36** | **39.50** |

adapters are attached only to the CA layers. This configuration significantly reduces the FD score to 0.047, increases semantic similarity (CLIP$_{sim}$ = 0.36), and enhances image quality with a BRISQUE score of 39.50. These highlights that focusing the adapters on the cross-attention layers leads to the most substantial improvements in both fairness and image quality.

**Impact of orthogonal regularization (OR).** Table 6 demonstrates the impact of orthogonal regularization (OR). Without OR, the FD score is 0.143, semantic similarity (CLIP$_{sim}$) drops to 0.29, and image quality (BRISQUE) is 37.92. When OR is applied, perfor-

Table 6: Orthogonal Ablation

| Location | FD $\downarrow$ | CLIP$_{sim}$ $\uparrow$ | BRISQUE $\uparrow$ |
|---|---|---|---|
| Ours $w \backslash o$ OR | 0.143 | 0.29 | 37.92 |
| Ours $w \backslash$ OR | **0.047** | **0.36** | **39.50** |

mance improves significantly across all metrics, with a much lower FD score of 0.047, higher semantic similarity of 0.36, and improved image quality (BRISQUE = 39.50). This highlights the effectiveness of orthogonal regularization in reducing bias and improving overall performance.

## 6 CONCLUSION

In this paper, we propose DebiasDiff, a plug-and-play method that learns attribute latent directions in a self-discovering manner, thus mitigates the reliance on collecting additional reference datasets. Our method can not only jointly debias multiple attributes in DMs, but also enables the generated images to follow a prescribed attribute distribution. It is lightweight and can be integrated with other DMs without re-training. Extensive experiments on debiasing gender, racial, and their intersectional biases show that our method outperforms previous SOTA by a large margin. We believe that our work marks a critical advancement in addressing harmful societal stereotypes within diffusion models, and it contributes to the ethical real-world applications of text-to-image diffusion models.

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

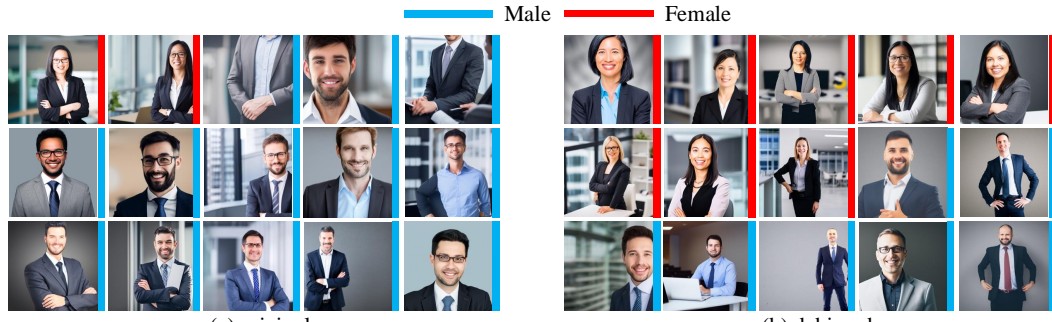

(a) original                                                    (b) debiased

Figure 8: Images generated from the original SD (left) and Ours for gender and race (right) with prompt 'A photo of a ceo'. Gendet ratio: Male : Female = 13 : 2 → 7 : 8

# A  APPENDIX

## A.1  IMPLEMENTATION DETAILS

We use Stable Diffusion v2.1 for all methods. We employ the prompt template "*a photo of the face of a* {occupation}*, a person*". At inference time, for each bias, we generate 100 images per occupation across 100 occupations, resulting in a total of 10,000 images. We set $\eta = \alpha = 1$, and train for 1000 iterations with a learning rate of 1e-5. For gender bias, we use the CelebA (Liu et al., 2015) dataset to train a binary classifier with two categories:{male,female}. For racial bias, we use the FairFace (Joo, 2021) dataset to train a classifier with the following four categories: WMELH={White, Middle Eastern, Latino Hispanic}, Asian={East Asian, Southeast Asian}, Black, and Indian. Please refer to supplementary for more details. We also conduct experiments with other versions of Stable Diffusion.

## A.2  COMPARED METHODS

**Finetuning for Fairness (F4Fair)** (Shen et al., 2024) is a training-free approach with two main technical innovations: (1) a distributional alignment loss that aligns specific attributes of generated images to a user-defined target distribution, and (2) adjusted direct finetuning (adjusted DFT) of the diffusion model's sampling process, which uses an adjusted gradient to directly optimize losses on generated images.

**H-Distribution Guidance (H Guidance)** (Parihar et al., 2024) is another training-free approach. It introduces *Distribution Guidance*, which ensures that generated images follow a prescribed attribute distribution. This is achieved by leveraging the latent features of the denoising UNet, which contain rich demographic semantics, to guide debiased generation. They also train an *Attribute Distribution Predictor* (ADP), a small MLP that maps latent features to attribute distributions. ADP is trained using pseudo labels generated by existing attribute classifiers, allowing fairer generation with the proposed Distribution Guidance.

**Unified Concept Editing (UCE)** (Gandikota et al., 2023) is a closed-form parameter-editing method that enables the application of numerous editorial modifications within a single text-to-image synthesis model, while maintaining the model's generative quality for unedited concepts.

## A.3  MORE VISUALIZATION RESULTS

We provide more visualization results about gender debaising and racial debaising. The qualitative results in Figure 8 9 10 further demonstrate that our method(DebiasDiff) effectively mitigates gender bias without compromising image quality or semantic coherence.

The qualitative results in Figure 11 12 further verify that our method outperforms others in reducing racial bias while preserving both semantic similarity and image quality.

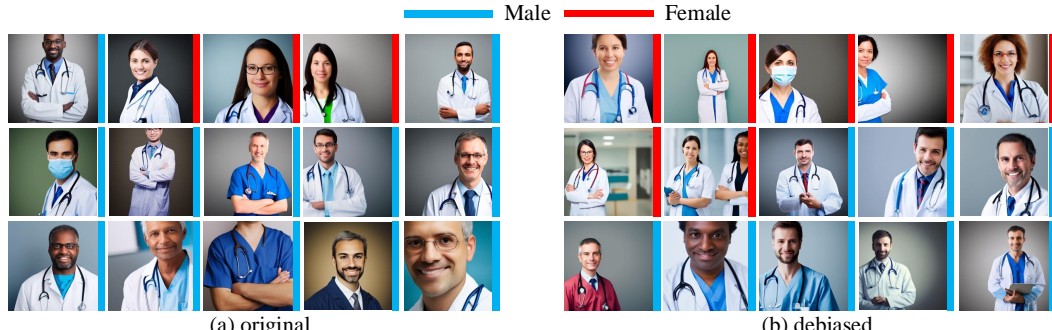

Figure 9: Images generated from the original SD (left) and Ours for gender and race (right) with prompt 'A photo of a doctor'. Gendet ratio: Male : Female = 12 : 3 → 8 : 7

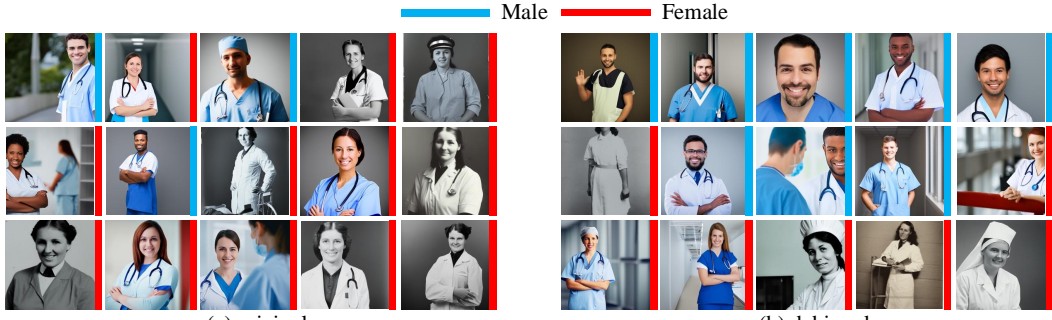

Figure 10: Images generated from the original SD (left) and Ours for gender and race (right) with prompt 'A photo of a nusrse'. Gendet ratio: Male : Female = 3 : 12 → 7 : 8

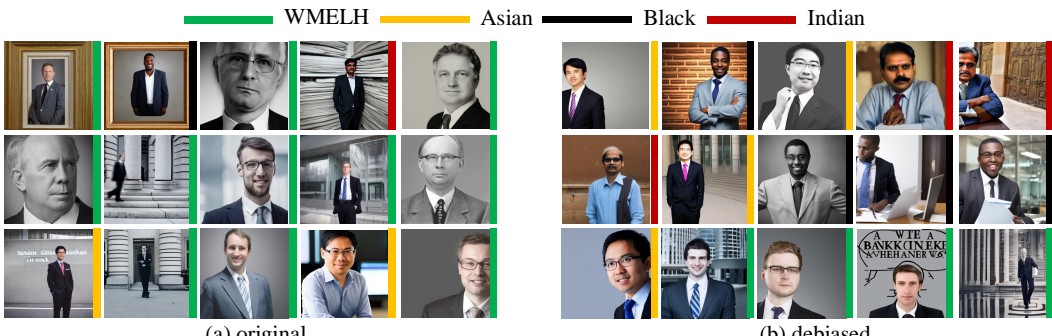

Figure 11: Images generated from the original SD (left) and Ours for gender and race (right) with prompt 'A photo of a banker'. Racial group distribution: WMELH : Asian : Black:Indian = 10:2:1:1 → 4:4:4:3

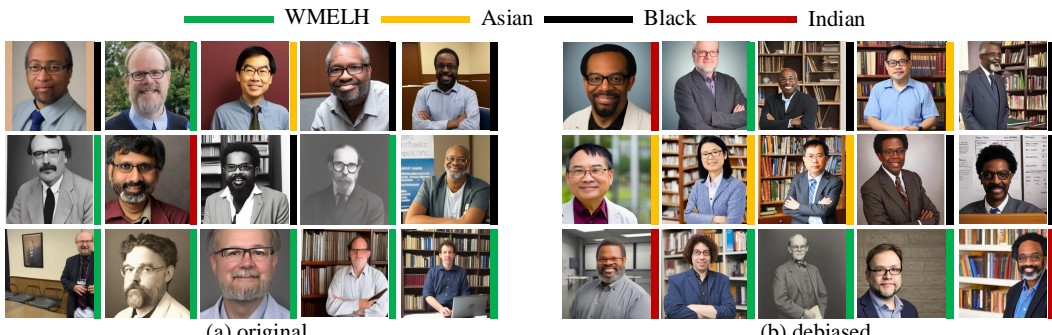

Figure 12: Images generated from the original SD (left) and Ours for gender and race (right) with prompt 'A photo of a professor'. Racial group distribution: WMELH : Asian : Black:Indian = 8:1:5:1 → 4:4:4:3

