# OpenReview forum: "Debiasing Text-to-image Diffusion Models with Self-discovering Latent Directions"
_ICLR.cc/2025/Conference — ICLR 2025 Conference Withdrawn Submission_

### Official Review · Reviewer_Qc64 · 2024-11-01

**Soundness:** 3
**Presentation:** 4
**Contribution:** 1
**Rating:** 3
**Confidence:** 5

**Summary:**

This paper proposes to debias diffusion models without retraining the model or using addition data or attribute classifiers. For each attribute, each category/class direction is learnt in a self-discovering manner by training 1-Dimensional Adapters attached to the cross attention blocks of the Unet. Once these category directions are learnt, they are use to debias the model during inference by using a distribution indicator which suggests which adapters to use for each image to obtain the required attribute distribution.

**Strengths:**

1. The paper is well written, and it is easy to follow.
2. The proposed approach does not need model retraining or full finetuning. Further, no additional data or classifiers are required.
3. The authors have compared the results with relevant baselines.
4. Transferability of the trained direction adapters across various stable diffusion variants is showcased.

**Weaknesses:**

1. (Major) Novelty:

In the model, it seems like the main novelty comes from the self-discovering training process of the adapters for each attribute category. However [1] already explores the idea of creating category directions for attributes. [1] discovers attribute directions by fine-tuning LoRA adaptors on a diffusion model with the loss: $\epsilon L = \epsilon\theta(X, c_t, t) + \eta(\epsilon\theta(X, c_+, t) - \epsilon\theta(X, c_-, t))$

If the negative prompt $c_-$ is replaced by a neutral prompt $c_t$, it becomes the identical to the loss used in this paper. So, in terms of the loss formulation, it is similar to [1]. From an architectural perspective, while [1] uses LoRA, this work uses 1-D adapters (as used in [2]), different approaches both aiming to alleviate full fine-tuning of the DM.
The modification from [2] seems to be that it is deploying the adapters in only the cross-attention layers rather than the whole Unet, which as a stand-alone does not amount to much novelty.
---

2. Intersectional Debiasing:

The paper employs an orthogonal loss between attributes when tackling multi-attribute debiasing. This dependency during training hampers the reusability aspect of these adapter. Ex: If I were to individually debias gender, race and age, I would train adapter directions for each. But now if I want to debias 2 of them, or all 3 attributes at a time, I cannot reuse these learnt single attribute direction vectors. This problem becomes more apparent especially in the case of intersection of multiple attributes.

Although I do understand that the orthogonality loss during training improves the numbers (as shown in the ablations section), it would be interesting to explore how these models trained for individual attribute directions can be leveraged for multi-attribute debiasing, without needing to retrain them for each subset of attributes.

---
3. (Minor) Architectural Diagram: The equation in Figure 4 has an error that needs correction:

$\epsilon\theta(X_t, g_t, t) + \eta(\epsilon\theta(X_t,$  **d**$, t) - \epsilon\theta(X_t, g_t, t))$

---
---
References:

[1] Rohit Gandikota, Joanna Materzyńska, Tingrui Zhou, Antonio Torralba, David Bau. "Concept Sliders: LoRA Adaptors for Precise Control in Diffusion Models" arXiv preprint arXiv:2311.12092 (2023).

[2]  Rishubh Parihar, Abhijnya Bhat, Abhipsa Basu, Saswat Mallick, Jogendra Nath Kundu, and R.Venkatesh Babu. Balancing act: Distribution-guided debiasing in diffusion models, 2024. https://arxiv.org/abs/2402.18206

[3]  Mengyao Lyu,Yuhong Yang, Haiwen Hong, Hui Chen, Xuan Jin,Yuan He, Hui Xue, Jungong Han, and Guiguang Ding. One-dimensional adapter to rule them all: Concepts, diffusion models and erasing applications, 2024. https://arxiv.org/abs/2312.16145.

**Questions:**

1. How is this work different from [1] in terms of what it brings to the table?
2. How does it compare with [1] in terms of metrics, compute time, etc?
3. In terms of comparison against baselines, [3] uses FID to evaluate the image quality, while the authors mention that they use BRISQUE as in the prior work [3].
4. The FD numbers reported in [3] are much lower than that reported in this paper. What is the reason for this discrepancy? Upon delving further into [3] I noticed that [3] has provided the code for unconditional diffusion models, while this paper talks about using stable diffusion in the experiments section (line 612), how is this difference accounted for?


---
Please correct me if you think I have misunderstood any aspect of the paper

---

### Official Review · Reviewer_Tq89 · 2024-11-04

**Soundness:** 3
**Presentation:** 3
**Contribution:** 2
**Rating:** 5
**Confidence:** 3

**Summary:**

The paper introduces a method for debiasing models without relying on external reference datasets. It employs self-discovered attribute adapters, which are optimized in alignment with a latent direction. These adapters are then adjusted to fit a target distribution using a distributional indicator, showing improved results over previous state-of-the-art approaches.

**Strengths:**

1. The paper presents learning adapters for various attributes, such as gender and race, in a self-discovering manner.
2. This method operates without requiring additional annotated reference datasets, which simplifies the process.
3. The results indicate improvement compared to previous state-of-the-art methods, with ablation studies supporting key architectural choices.

**Weaknesses:**

1. The optimization process for learning adapters closely resembles [1], potentially limiting the method’s novelty.
2. The study omits comparisons with prior self-discovering debiasing methods, such as [2], which learns a latent vector for each concept. This further limits the uniqueness of the contribution.
3. Comparison protocols differ from previous works ([1], [4]), omitting the Winobias dataset ([3]). Justifying this deviation would clarify the appropriateness of the comparisons.


[1] Concept Sliders: LoRA Adaptors for Precise Control in Diffusion Models (ECCV 2024).
[2] Self-Discovering Interpretable Diffusion Latent Directions for Responsible Text-to-Image Generation (CVPR 2024).
[3] Gender Bias in Coreference Resolution: Evaluation and Debiasing Methods (NAACL 2018).
[4] Unified Concept Editing in Diffusion Models (WACV 2024).
[5] Finetuning Text-to-Image Diffusion Models for Fairness (ICLR 2023).

**Questions:**

1. The loss function for LoRA adaptation appears similar to that in [1], yet it wasn't cited. Could you elaborate on how your formulation differs from that work?
2. The standard Winobias dataset [3] wasn't included in your comparison. Could you share your rationale for omitting it?
3. How does your approach diverge from previous self-discovery methods, and how do the evaluation results compare?
4. According to generation ratio (e.g., female and male) adapter corresponding to particular attribute need to be selected e.g., adapter for female or male internally. Why not simply select the token internally corresponding to the attribute and pass it along the text on which the generate image in conditioned.

---

### Official Review · Reviewer_K3z1 · 2024-11-04

**Soundness:** 3
**Presentation:** 3
**Contribution:** 3
**Rating:** 6
**Confidence:** 4

**Summary:**

The paper is about debiasing diffusion models using a combination of attribute adapters and a distribution indicator . The paper proposes a novel technique for training the attribute adapters without any additional data and uses the distribution indicator with the attribute specific adapters to debias the diffusion model . The paper demonstrates strong performance across different attributes and datasets while maintaining the generation quality of the original diffusion model.

**Strengths:**

* Training the attribute specific classifiers for finding directions in the latent space does
not require additional training data.

* Applying distribution indicator during inference is computationally cheaper and faster
instead of using the distribution to train the attribute classifiers in the latent space .

* Strong results on single and multiple attributes .

* The technique can be transferred to multiple diffusion models without retraining multiple
times .

* Ablation studies show each component is important for the final performance .

**Weaknesses:**

* The results are shown on limited types of biases (only gender and race). Evidence that the method for other kinds of biases might be helpful (see questions).

* All results are shown single person or single occupation. What if multiple people are generated for a single occupation?

* All results are shown for a fixed ratio of bias (i.e, generations are balanced across the chosen bias attribute). Ablations should be there for different ratios as shown in ITI-Gen [a]

* Comparisons with competitive methods are less (e.g.,[a, b]).


        [a] Zhang, Cheng, et al. "Iti-gen: Inclusive text-to-image generation." Proceedings of the IEEE/CVF International Conference on Computer Vision. 2023.
       [b] Li, Hang, et al. "Self-discovering interpretable diffusion latent directions for responsible text-to-image generation." Proceedings of the IEEE/CVF Conference on Computer Vision and Pattern Recognition. 2024.

**Questions:**

1. Can you show some results for more type of biases (hair color , eyeglasses etc) ?
2. How does the technique perform for scene based models or images with multiple persons (say 2/3)?
3. Can you show some results for different types of ratio for a specific bias ? Example for gender (Male, Female) can you show results from where the ratio of Male:Female goes from (0:100) to (100:0) ?

---

### Note · Authors · 2024-11-15

**Comment:**

We appreciate the reviewer for their feedbacks.

**Withdrawal Confirmation:**

I have read and agree with the venue's withdrawal policy on behalf of myself and my co-authors.